# Microbiota Metabolism Failure as a Risk Factor for Postoperative Complications after Aortic Prosthetics

**DOI:** 10.3390/biomedicines11051335

**Published:** 2023-04-30

**Authors:** Natalia Beloborodova, Alisa Pautova, Marina Grekova, Mikhail Yadgarov, Oksana Grin, Alexander Eremenko, Maxim Babaev

**Affiliations:** 1Federal Research and Clinical Center of Intensive Care Medicine and Rehabilitology, 25-2 Petrovka Str., 107031 Moscow, Russia; alicepau@mail.ru (A.P.); mikhail.yadgarov@mail.ru (M.Y.); 2Petrovsky Russian Research Center of Surgery, 2 Abrikosovsky Pereulok, 119991 Moscow, Russia; levitskayams@yandex.ru (M.G.); grin_oksana@mail.ru (O.G.); aeremenko54@mail.ru (A.E.); maxbabaev@mail.ru (M.B.)

**Keywords:** cardiovascular surgery patients, aortic aneurism, aortic dissection, aromatic microbial metabolites, tyrosine metabolites, 4-hydroxyphenyllactic acid, infectious complications

## Abstract

Postoperative complications in cardiovascular surgery remain an important unresolved problem, in particular in patients with aortic aneurysm. The role of the altered microbiota in such patients is of great interest. The aim of this pilot study was to determine whether the development of postoperative complications in patients with aortic aneurysm is related with initial or acquired disorders of microbiota metabolism by monitoring the level of some aromatic microbial metabolites (AMMs) circulating in the blood before the surgery and in the early postoperative period. The study comprised patients with aortic aneurysm (*n* = 79), including patients without complications (*n* = 36) and patients with all types of complications (*n* = 43). The serum samples from the patients were collected before and 6 h after the end of the surgery. The most significant results were obtained for the sum of three sepsis-associated AMMs. This level was higher before the surgery in comparison with that of healthy volunteers (*n* = 48), *p* < 0.001, and it was also higher in the early postoperative period in patients with all types of complications compared to those without complications, *p* = 0.001; the area under the ROC curve, the cut-off value, and the odds ratio were 0.7; 2.9 µmol/L, and 5.5, respectively. Impaired microbiota metabolism is important in the development of complications after complex reconstructive aortic surgery, which is the basis for the search for a new prevention strategy.

## 1. Introduction

With major reconstructive operations in cardiovascular surgery, the problem of the development of postoperative complications remains extremely relevant, since complications affect the short- and long-term results of surgical intervention, and also significantly increase financial costs. Aortic aneurysm is a severe life-threatening disease, which occurs with the involvement of almost all systems and organs in the pathological process, is characterized by hypoperfusion of organs, impaired tissue metabolism and progression of the inflammatory process, and is often accompanied by aortic dissection with the threat of fatal bleeding, etc. [1,2]. In modern cardiac surgery clinics, it is possible to minimize the adverse effect of a number of intraoperative factors during aortic prosthetics, namely to reduce the time of cardiopulmonary bypass and surgical intervention, the volume of blood loss, the degree of patient’s cooling, etc. Excellent results have recently been achieved without lethal outcomes, even such large-scale operations as replacement of the entire aorta in patients with widespread aneurysmal dilatation of the aorta [3] or hybrid aortic repair in patients with type III aortic dissection and concomitant proximal aortic lesion [4]. Despite the fact that sterility is guaranteed in operating rooms and that perioperative antibiotic prophylaxis is widely used, the level of local and systemic infectious complications, including septic conditions, is still high. In a significant number of patients, complications develop not only in the early postoperative period, but also after discharge from the surgical clinic. For example, aortic endograft infection after endovascular aneurysm repair is one of the most dangerous infectious complications with the lethality of 16.9–39.2% [5].

The working hypothesis of this study is that microbiota disorders initially accompany aortic aneurysm and may subsequently cause the development of postoperative complications in conditions when preoperative preparation, surgery and early intensive therapy do not contribute to improvement, but, on the contrary, aggravate previous disorders.

In choosing research methods, the authors relied on a number of previous works. The fact of deep disturbances in the composition of the gut microbiota was revealed during the examination of patients in intensive care units using the 16S rRNA gene sequencing method. The authors point to the loss of biodiversity, a sharp reduction in the number of species of beneficial anaerobic bacteria inherent in the body of a healthy person, and excessive growth of potentially pathogenic microorganisms [6,7,8,9]. The mechanism for the development of complications associated with the altered microbiota of a cardiac surgery patient was also studied. It was reported that in planned heart operations (heart valve surgery, coronary artery bypass grafting, etc.), postoperative infectious complications were observed in 17% of cases, and the species composition of the microbiota in the group of patients with infectious complications was significantly changed initially in comparison with the group without complications [10].

Changes in the profile of microbiota metabolites in the gut and then in the blood are secondary to species dysbiosis. At the same time, the assessment of the degree of microbiota metabolism disorder may be clinically more significant than information about taxonomic disorders [11,12]. The chapter in the work by Beloborodova N.V., 2022 summarizes the results of the study of microbial metabolites of aromatic structure as markers for monitoring septic conditions in intensive care units [13]. Earlier, an explanation was offered describing an increase in serum concentrations of aromatic amino acid metabolites with its high diagnostic significance. The accumulation of intermediates of tyrosine and phenylalanine metabolism, namely phenyllactic, 4-hydroxyphenylacetic and 4-hydroxyphenyllactic acids, is a consequence of a violation of the microbiota, which loses the ability to biodegrade aromatic amino acids [14]. Later, the bacterial metabolic pathway of aromatic amino acids was studied in more detail and, using the model of gnotobiological mice, aromatic microbial metabolites was shown to affect intestinal permeability and systemic immunity [15]. Thus, the profile and levels of a number of aromatic amino acid metabolites circulating in the blood are associated with taxonomic and metabolic disorders of the gut microbiota, which can be used to assess the degree of metabolic dysfunction of the microbiota. In cardiac diseases, the study of microbiota metabolites is of interest both as markers of postoperative sepsis [16] and as participants in the genesis of myocardial infarction [17].

It is always difficult to compare groups of patients and the results of surgical treatment, especially in regard to such pathology as aortic aneurysm where different types of operations are used and much depends on the experience of surgeons and anesthesiologists. In order to avoid doubts about the results that could be related to the qualifications and experience of medical personnel, the ethics committee made a decision to include in this study patients operated on by one specific team of surgeons and treated by a specific team of specialists in anesthesia and intensive care.

The aim of this study was to determine whether the development of postoperative complications in patients with aortic aneurysm is related with initial or acquired disorders of microbiota metabolism by monitoring the level of aromatic microbial metabolites (AMMs) circulating in the blood before the surgery and in the early postoperative period (6 h).

## 2. Materials and Methods

### 2.1. Study Design

The present study was a single-center clinical study performed at the Petrovsky Russian Research Center of Surgery (Moscow, Russia). Patients with a diagnosis of aneurysm/aortic dissection were included in the study. In order to exclude the influence of individual qualities and skill of surgeons and anesthesiologists, all patients involved in the study were operated on by the same team of surgeons and anesthesiologists.

Inclusion criteria: patients aged from 18 to 75 years; reconstructive aortic surgery in 2021 performed by a team of surgeons under the guidance of Corresponding Member of the Russian Academy of Sciences, Professor, MD E.R. Charchyan; and a team of anesthesiologists under the guidance of Professor of the Russian Academy of Sciences, MD B.A. Axelrod; the presence of informed consent statement.

Exclusion criteria: age under 18 or over 75 years; aortic surgery performed by another surgical team; refusal of the patient to participate in the study at any stage; mental disorders that prevent obtaining informed consent statement; and patient’s transfer to another hospital early after surgery.

### 2.2. Biological Samples

The total number of the serum samples was 206, including 158 samples from the patients and 48 samples from the healthy volunteers. The samples of blood serum from the patients (*n* = 79) before the surgery and 6 h after the end of the surgery were collected and frozen at −30 °C in Petrovsky Russian Research Center of Surgery (Moscow, Russia). The Local Ethic Committee approved the study (N7 from 15 February 2021) which was conducted in accordance with the ethical standards of the Declaration of Helsinki and formal consent for participation in this study was obtained from each patient or their legal representative. The samples of the blood serum from the healthy volunteers (*n* = 48) were collected in Federal State Budgetary Institution N.N. Burdenko Main Military Clinical Hospital (Moscow, Russia). The concentrations of aromatic acids (benzoic, phenylpropionic, phenyllactic, 4-hydroxybenzoic, 4-hydroxyphenylacetic, 4-hydroxyphenylpropionic, homovanillic, and 4-hydroxyphenyllactic acids) were measured using gas chromatography-mass spectrometry (Trace GC 1310 gas chromatograph and ISQ LT mass spectrometer, Thermo Electron Corporation, Waltham, MA, USA) in Federal Research and Clinical Center of Intensive Care Medicine and Rehabilitology (Moscow, Russia) as it was previously described [18]. The limit of quantitation for all aromatic acids is 0.5 µmol/L with relative standard deviation of 10–30%. The calibration curves were linear functions for all aromatic metabolites in the most clinically significant range of concentrations 0.5–7 µmol/L.

### 2.3. Statistical Analysis

The Shapiro–Wilk test was used to assess the normality of the data distribution. Continuous and categorial variables were described using median and interquartile ranges, frequency and percentages, respectively. Group differences were explored using Mann–Whitney U test for continuous variables; categorical baseline variables were analyzed using Chi-square test and Fisher’s exact test. The cut-off value was chosen using the receiver operating characteristic (ROC) analysis with the assessment of area under the curve parameter and its 95% confidence intervals (CI) in order to achieve the optimal sensitivity/specificity ratio (Youden’s J statistic). Additionally, odds ratio, sensitivity, specificity, positive and negative predictive values and accuracy were calculated. CI for sensitivity and specificity are “exact” Clopper–Pearson CI. Scatter plots with linear regression were used for visualization and modeling the relationship between metabolic status of the patients and anthropometric parameters. All analyses were conducted using IBM SPSS Statistics for Windows, Version 27.0. Armonk, NY, USA: IBM Corp. The differences were considered significant at *p* < 0.05.

## 3. Results

### 3.1. Patients

From 166 patients with thoraco-abdominal aorta surgeries performed in 2021, 81 patients met the protocol conditions because they were operated on by one specific team. Two patients were excluded from the analysis due to the positive test for COVID-19 infection and were transferred to another clinic. Finally, 79 patients were included in the study. The median age of patients was 57 (46–64) years, 57 (72%) men and 22 (28%) women. There were 15 (19%) patients with acute/subacute aortic dissection (Figure 1). All patients survived.

Several concomitant diseases were diagnosed in all patients participating in the study, including hypertension in 57 (72%) and heart defects in 55 (70%) patients, as well as various diseases of the gastrointestinal tract in 48 (61%) patients, multifocal atherosclerosis in 24 (30%) patients, ischemic heart disease in 20 (25%) patients, rhythm disturbances and cardiac conduction in 18 (22%) patients, and chronic kidney disease in 16 (20%) patients (Appendix A). Prognostic comorbidity assessed by Charlson comorbidity index [19] was 4 (2; 5).

The patients underwent different types of reconstructive operations on the aorta, including

Prosthetics of one or more parts of the thoracic aorta, *n* = 21 (27%);Hybrid surgery: stenting of the descending thoracic aorta with or without plastic surgery/prosthetics of the root and ascending aorta using Bentall–DeBono techniques or T. David, *n* = 25 (32%);Prosthetics of the aortic valve and ascending aorta using the Bentall–DeBono technique, *n* = 14 (18%);Prosthetics of the root and ascending aorta by the method of David, *n* = 9 (11%);Prosthetics of the thoracoabdominal aorta, *n* = 5 (6%).

Some patients underwent additional surgical interventions at the same time:Prosthetics/plastics of the aortic, mitral or tricuspid valve, *n* = 19 (24%);Myocardial revascularization (aorto-mammary coronary, prosthetic coronary bypass surgery), *n* = 12 (15%);Radiofrequency ablation, *n* = 3 (4%).

For perioperative antibiotic prophylaxis in 57 (72%) patients, cefazolin was standardly used (2 g once immediately before surgery), and then it was prescribed at a dosage of 2 g three times a day for two or three days. Other cephalosporins (cefuroxime, ceftazidime) or antimicrobials of other groups or their combinations with vancomycin, fluconazole, etc., were rarely used according to individual indications, taking into account the anamnesis. When comparing indicators in groups of patients with and without complications who received different antimicrobials for antibiotic prophylaxis, no differences were detected.

### 3.2. Aromatic Metabolites in Patients and Healthy Volunteers

A number of tyrosine and phenylalanine metabolites was measured in the serum samples of the patients and healthy volunteers, and the limit of quantitation for all metabolites was 0.5 µmol/L (Table 1). The serum samples of all patients before the surgery (*n* = 79) were characterized by higher values of benzoic and 4-hydroxyphenyllactic acids compared to the healthy volunteers (*n* = 48). The parameter which describes the content of the diagnostically significant sepsis-associated microbial metabolites [20] includes the sum of phenyllactic, 4-hydroxyphenylacetic and 4-hydroxyphenyllactic acids (Σ3AMM). This parameter was also higher in the serum samples of the patients before the surgery. The concentration of other aromatic acids (phenylpropionic, phenyllactic, 4-hydroxybenzoic, 4-hydroxyphenylacetic, 4-hydroxyphenylpropionic, and homovanillic acids) was less than the limit of quantitation (0.5 µmol/L) in most cases.

### 3.3. Aromatic Metabolites in Different Groups of Patients

All patients (*n* = 79) were divided into groups according to the course of the postoperative period. There were patients without complications (*n* = 36) and patients with different types of complications (*n* = 43). More than half of patients with a complicated course of the postoperative period had infectious complications (*n* = 26) such as pneumonia or local infection of the skin and soft tissues in the area of surgery. Among other types of complications, there were the development of organ dysfunctions or surgical complications (bleeding, hematomas in the area of surgery), etc. Most often, patients with complications needed longer treatment in the ICU or were re-transferred from the specialized department to the intensive care at different times of the postoperative period, for example, due to the development of respiratory or cardiovascular insufficiency.

The serum samples from the patients were collected before the surgery (0 points) and 6 h after the end of the surgery (1 point). The characteristics of the patients, the intraoperative parameters, the length of stay, and the concentrations of aromatic metabolites are demonstrated in Table 2. 

The intraoperative parameters (time of cardiopulmonary bypass and myocardial ischemia, drainage, intraoperative and total blood loss) and length of hospital stay were statistically different in the patients with all types of complications (*n* = 43) and without complications (*n* = 36). The concentration of one of the sepsis-associated metabolites 4-hydroxyphenyllactic acid in serum sample collected 6 h after the end of surgery (1 point) and the difference (∆ 1–0) was higher in patients with complications (*n* = 43). The same results were obtained for the sum of sepsis-associated microbial metabolites Σ3AMM.

The results for the comparison of the patients without complications (*n* = 36) and the patients with infectious complications (*n* = 26) were similar to those for patients with all types of complications (*n* = 43).

Age, gender, metabolic status of the patients (BMI), concomitant diseases and other initial factors of patients, as well as the general index of concomitant pathology of Charlson had no significant differences between the two groups of patients. We detected a trend towards an increase in Σ3AMM with increasing age that did not reach statistical significance (*p* = 0.065), and there was no significant relationship between BMI and Σ3AMM (*p* = 0.892) (Appendix A).

### 3.4. Aromatic Metabolites for the Assessment of the Risk of Postoperative Complications in Patients

The ROC-analysis (Figure 2, Table 3) was carried out for the 4-hydroxyphenyllactic acid and Σ3AMM concentrations in serum samples collected 6 h after the end of the surgery, which were significantly different in the patients with all types of complications (*n* = 43) and without complications (*n* = 36). The areas under the ROC curve for 4-hydroxyphenyllactic acid and Σ3AMM were 0.686 and 0.717, respectively. The optimal cut-off values for 4-hydroxyphenyl lactic acid (2.0 µmol/L) and Σ3AMM (2.9 µmol/L) were calculated. The risk of postoperative complications was 3.4 times greater (95% CI 1.3–9.0) and 5.5 times greater (95% CI 1.9–15.0) for 4-hydroxyphenyl lactic acid and Σ3AMM, respectively, when the cut-off value was exceeded.

## 4. Discussion

Despite the great progress in cardiovascular surgery, the percentage of postoperative complications, such as infections at the site of surgery, pneumonia, and others, does not decrease and, according to the literature, on average is registered to range from 12.6 to 21%. The studies on this topic provide analyses of the risk factors for complications, which most often include age, gender, chronic lung diseases, heart failure, duration of cardiopulmonary bypass [21,22,23]. In recent years, there has been a growing interest in studying the potential connection of human microbiota with cardiovascular diseases [24,25] and the role of the intestinal microbiota in the pathogenesis of heart failure [26,27]. Scientific data confirm the potential involvement of microbiota-related factors, for example, taxonomy features of gut microbiota in cardiosurgical intensive care patients [28] and increased risk of bacterial translocation followed by a pro-inflammatory condition [24,29]. The studies consider the possibilities of influencing the microbiota in order to improve the results of treatment of patients with heart diseases in the future [30,31,32]. However, we did not detect any studies describing the relationship between the metabolic activity of the microbiota though metabolites of tyrosine and phenylalanine and the development of cardiac surgical complications.

In this study, the authors took as a basis the postulate that “function is more important than systematics” [13] and assessed the state of the microbiota by the products of microbial metabolism of aromatic amino acids circulating in the blood. The obtained results confirmed the working hypothesis that patients with aortic aneurysm had deviations in the level of circulating AMMs at the stage of admission to the clinic, before surgery, in comparison with the healthy volunteers. This fact, on the one hand, is a manifestation of microbiota dysfunction as part of multiple organ disorders in conditions of atherosclerotic process, hypertensive and ischemic diseases, and genetically determined connective tissue dysplasia [33]. On the other hand, a disorder of the integrity of the vascular wall against the background of an inflammatory process of various etiologies may be due to active changes in microbial metabolism and bacterial translocation. Risk factors for the development of perioperative complications (surgical trauma, prolonged cardiopulmonary bypass, myocardial ischemia, massive blood loss) additionally disrupt the metabolic function of the microbiota, which was manifested by a statistically significant increase in serum AMMs 6 h after surgery (1 point) compared to those before surgery (0 points) in patients with complications (Table 2).

Age, gender, BMI, concomitant diseases and other initial factors of patients, as well as the general index of concomitant pathology of Charlson displayed no significant differences between the two groups of patients. At the same time, aromatic metabolites, such as 4-hydroxyphenylacetic and 4-hydroxyphenyllactic, and especially the sum of three AMMs, were significantly higher (*p* < 0.05). In the group of patients with complications, the median value of the difference of the sum of three AMMs (∆ 1–0) was more than two times higher than this level in the group without complications (*p* = 0.001), which can be used for monitoring and prognosis of complications in the future. It is noteworthy that 6 h after the surgery, the sum of three AMMs in patients with infectious complications (*n* = 26) was within 4.3 (3.0–5.7) µmol/L and did not reach high values (up to 10 µmol/L or more), as, for example, it was reported in patients with sepsis [20,34]. Moreover, all patients survived, and the average duration of stay of patients in the intensive care unit was 1 day for patients without complications in comparison with 4 days for patients with all types of complications (*p* < 0.001).

It is also interesting to note that not all patients at the highest risk with acute/subacute aortic dissection (*n* = 15) were in the group with complications after surgery. As can be seen from Figure 1, four patients (27%) had no postoperative complications. This fact once again indicates the high significance of microbiota metabolic disorders compared, for example, to the fact of the acute aortic dissection or the scope of surgical intervention.

Our study is a pilot single center study with obvious limitations. In the medical center where the study was conducted, 1087 different cardiac surgical interventions were performed in 2021, including 166 (15%) surgeries for aortic aneurysm/dissection. In accordance with the protocol of the study, patients (*n* = 79) which were operated on by one surgical team in 2021 were analyzed in our study. We purposefully selected operations on the aorta due to the greatest risk of postoperative complications, taking into account the severity of this pathology and a complex of concomitant diseases in every patient included in the study (Appendix A). All patients with this pathology need individually selected options for reconstructive aortic surgery, differing in type, localization, duration, and volume, which are listed in Section 3.1. Despite the small number of patients, a disorder in metabolic function of the microbiota as an important risk factor for postoperative complications was identified, which was not previously taken into account. In addition, it was determined that this risk factor can be determined with a high degree of confidence early enough after surgery (6 h). The authors assume that this important result will be confirmed with other types of operations in further extensive studies. 

It is important to consider that operations on the aorta in such patients with multiple risk factors are not performed in all cardiac surgery clinics, which makes it difficult to organize multicenter studies, and it is almost impossible to select a group of similar patients.

The authors believe that the results of this pilot study are extremely important for understanding the mechanisms of complications. Impaired microbiota metabolism is important in the development of complications after complex reconstructive aortic surgery, which is the basis for the search for a new prevention strategy. At the time of writing of this paper, a randomized clinical trial has already begun in the same cardiac surgery center to study the effectiveness of antimicrobial prevention based on the regulation of the metabolic activity of the microbiota.

## 5. Conclusions

The microbiota metabolism disruption is an important risk factor for postoperative complications in cardiovascular surgery, in particular, after aortic prosthetics. The degree of microbiota metabolism disruption can be assessed by monitoring the level of some AMMs circulating in the blood. It is necessary to look for the ways to adjust the metabolism of the microbiota to improve the results of surgical treatment in future by searching for replacement the compensation of its functions.

## Figures and Tables

**Figure 1 biomedicines-11-01335-f001:**
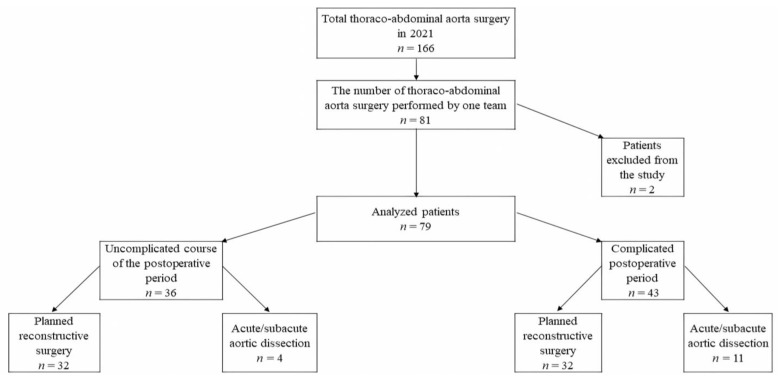
Scheme of the patients’ recruitment in a prospective single-center study who were admitted during 2021 for thoracoabdominal aortic prosthetics surgery.

**Figure 2 biomedicines-11-01335-f002:**
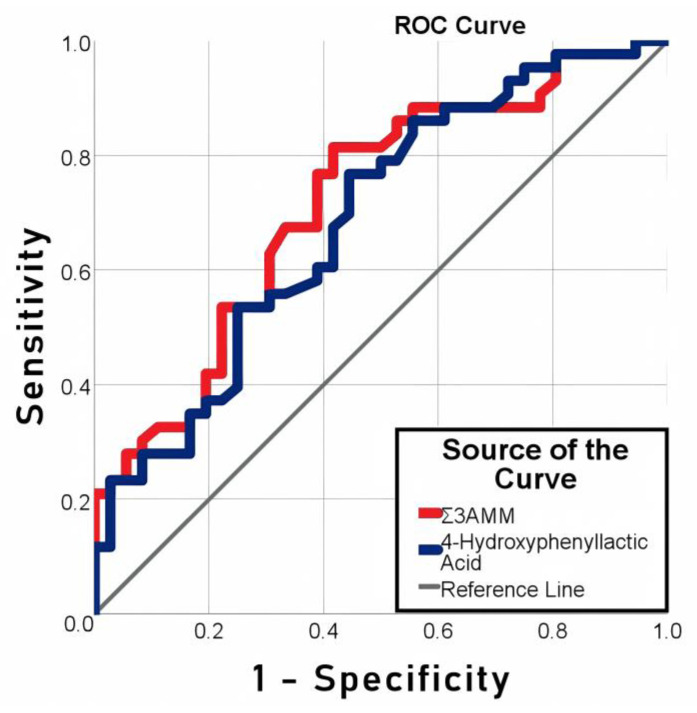
Prognostic value of 4-hydroxyphenyllactic acid and Σ3AMM in patients after cardiac surgery.

**Table 1 biomedicines-11-01335-t001:** The concentrations of aromatic metabolites in the serum samples of the patients before the surgery (*n* = 79) and healthy volunteers (*n* = 48), and the results of the Mann–Whitney U test.

Aromatic Acid, µmol/L	Healthy Volunteers (*n* = 48)	Patients (*n* = 79)	*p*-Value
Benzoic	0.5 (0.5–0.6)	1.2 (0.9–1.5)	<0.001
Phenylpropionic	<0.5 (<0.5–0.5)	<0.5 (<0.5–<0.5)	-
Phenyllactic	<0.5 (<0.5–<0.5)	<0.5 (<0.5–0.5)	-
4-Hydroxybenzoic	<0.5 (<0.5–<0.5)	<0.5 (<0.5–<0.5)	-
4-Hydroxyphenylacetic	<0.5 (<0.5–<0.5)	<0.5 (<0.5–0.6)	-
4-Hydroxyphenylpropionic	<0.5 (<0.5–<0.5)	<0.5 (<0.5–0.7)	-
Homovanillic	<0.5 (<0.5–<0.5)	<0.5 (<0.5–<0.5)	-
4-Hydroxyphenyllactic	1.3 (1.0–1.6)	1.6 (1.2–1.9)	0.003
Σ3AMM	1.9 (1.4–2.3)	2.3 (1.9–3.1)	<0.001

**Table 2 biomedicines-11-01335-t002:** The medical and demographic characteristics, the intraoperative parameters, the length of stay, and the concentrations of aromatic metabolites in the serum samples of the patients before the surgery (0 points), 6 h after the surgery (1 point) and the difference between these points (∆ 1–0) in all patients (*n* = 79), patients without complications (*n* = 36), patients with all types of complications (*n* = 43) including the patients with infectious complications (*n* = 26). The results of the Mann–Whitney U test for the patients without any complications (*n* = 36) and the patients with all types of complications (*n* = 43) are demonstrated as *p*-Value A; and for the patients without any complications (*n* = 36) and the patients with infectious complications (*n* = 26) are demonstrated as *p*-Value B. The statistically different data was highlighted with bold format.

Parameter	Patients (*n* = 79)	Patients without Complications (*n* = 36)	Patients with All Types of Complications (*n* = 43)	Patients with Infectious Complications (*n* = 26)	*p*-Value A	*p*-Value B
Medical and demographic characteristics						
Sex, male, %	57, 72.2%	25, 69.4%	32, 74.4%	22, 84.6%	0.6 *	0.2 *
Age, years	57 (46–64)	58 (45–63)	55 (48–66)	54 (48–66)	0.5	0.5
Body Mass Index (BMI), kg/m^2^	27.2 (24.2–30.7)	27.5 (23.9–31.2)	27.1 (24.1–30.0)	27.3 (24.8–30.4)	0.5	0.8
Charlson Comorbidity Index	4 (2–5)	3 (2–5)	4 (2–5)	4 (2–5)	0.3	0.3
Intraoperative parameters						
Acute/subacute aortic dissection	15, 81.0%	5, 13.9%	10, 23%	5, 19.2%	0.3 *	0.6 *
Cardiopulmonary Bypass, min	123 (101–162)	**111** (**73–140**)	**150** (**116–188**)	**163** (**138–217**)	**<0.001**	**<0.001**
Myocardial Ischemia, min	91 (66–117)	**82** (**55–102**)	**99** (**76–137**)	**119** (**92–142**)	**0.014**	**<0.001**
Intraoperative Blood Loss, mL	800 (700–1100)	**800** (**600–1000**)	**950** (**700–1500**)	**1000** (**800–2000**)	**0.019**	**0.01**
Drainage, mL	250 (150–370)	**190** (**110–300**)	**300** (**200–500**)	**325** (**200–550**)	**0.006**	**<0.001**
Total Blood Loss, mL	1100 (900–1500)	**1000** (**775–1300**)	**1250** (**960–1750**)	**1475** (**1020–2350**)	**0.003**	**<0.001**
Length of stay						
Length of Stay in the ICU	2 (1–4)	**1** (**1–1**)	**3** (**2–5**)	**4** (**2–10**)	**<0.001**	**<0.001**
Total Hospital Stay	10 (8–14)	**8** (**7–10**)	**13** (**10–18**)	**15** (**11–21**)	**<0.001**	**<0.001**
Aromatic acids, µmol/L						
Benzoic (0)	1.2 (0.9–1.5)	1.1 (0.9–1.4)	1.2 (1.0–1.6)	1.3 (1–1.5)	0.2	0.3
Benzoic (1)	1.3 (1.1–1.8)	1.3 (1.0–1.7)	1.2 (1.1–2.0)	1.2 (1.1–1.6)	0.7	0.9
∆ Benzoic (1–0)	0.1 (−0.3–0.6)	0.2 (−0.1–0.4)	0.1 (−0.4–0.7)	0 (−0.4–0.6)	0.7	0.5
Phenylpropionic (0)	<0.5 (<0.5–<0.5)	<0.5 (<0.5–<0.5)	<0.5 (<0.5–<0.5)	<0.5 (<0.5–<0.5)	-	-
Phenylpropionic (1)	<0.5 (<0.5–<0.5)	<0.5 (<0.5–<0.5)	<0.5 (<0.5–<0.5)	<0.5 (<0.5–<0.5)	-	-
∆ Phenylpropionic (1–0)	0 (−0.1–0)	−0.1 (−0.2–[−0.1])	0 (−0.1–0)	0 (−0.1–0)	0.2	0.3
Phenyllactic (0)	<0.5 (<0.5–0.5)	<0.5 (<0.5–0.5)	<0.5 (<0.5–0.5)	<0.5 (<0.5–<0.5)	-	-
Phenyllactic (1)	0.5 (<0.5–0.7)	0.5 (<0.5–0.6)	0.5 (<0.5–0.8)	0.6 (<0.5–0.8)	-	-
∆ Phenyllactic (1–0)	0.1 (0–0.3)	0.1 (0–0.2)	0.2 (0–0.3)	0.2 (0.1–0.3)	0.1	0.06
4-Hydroxybenzoic (0)	<0.5 (<0.5–<0.5)	<0.5 (<0.5–<0.5)	<0.5 (<0.5–<0.5)	<0.5 (<0.5–<0.5)	-	-
4-Hydroxybenzoic (1)	<0.5 (<0.5–<0.5)	<0.5 (<0.5–<0.5)	<0.5 (<0.5–<0.5)	<0.5 (<0.5–<0.5)	-	-
∆ 4-Hydroxybenzoic (1–0)	0 (0–0)	0 (0–0)	0 (0–0)	0 (0–0)	-	-
4-Hydroxyphenylacetic (0)	<0.5 (<0.5–0.6)	<0.5 (<0.5–0.6)	0.5 (<0.5–0.7)	0.5 (<0.5–0.8)	-	-
4-Hydroxyphenylacetic (1)	<0.5 (<0.5–0.7)	<0.5 (<0.5–<0.5)	0.5 (<0.5–1.1)	0.5 (<0.5–1.2)	**-**	**-**
∆ 4-Hydroxyphenylacetic (1–0)	0 (−0.2–0.2)	**−0.1** (**−0.2**–**0**)	**0.1** (**−0.1**–**0.4**)	**0.2** (**−0.1**–**0.5**)	**0.001**	**0.003**
4-Hydroxyphenylpropionic (0)	<0.5 (<0.5–0.7)	<0.5 (<0.5–0.7)	<0.5 (<0.5–0.5)	<0.5 (<0.5–<0.5)	-	-
4-Hydroxyphenylpropionic (1)	<0.5 (<0.5–0.7)	<0.5 (<0.5–0.7)	<0.5 (<0.5–0.6)	<0.5 (<0.5–<0.5)	-	-
∆ 4-Hydroxyphenylpropionic (1–0)	0 (0–0)	0 (0–0)	0 (0–0)	0 (0–0)	-	-
Homovanillic (0)	<0.5 (<0.5–<0.5)	<0.5 (<0.5–<0.5)	<0.5 (<0.5–<0.5)	<0.5 (<0.5–<0.5)	-	-
Homovanillic (1)	<0.5 (<0.5–<0.5)	<0.5 (<0.5–<0.5)	<0.5 (<0.5–0.8)	<0.5 (<0.5–<0.5)	-	-
∆ Homovanillic (1–0)	0 (0–0.3)	0 (0–0.1)	0.1 (0–0.6)	0 (0–0.3)	-	-
4-Hydroxyphenyllactic (0)	1.6 (1.2–1.9)	1.6 (1.2–1.9)	1.6 (1.2–1.9)	1.6 (1.2–1.9)	0.7	0.8
4-Hydroxyphenyllactic (1)	2.4 (1.8–3.2)	**2.0** (**1.6**–**2.9**)	**2.8** (**2.0**–**3.5**)	**2.8** (**2.0**–**3.7**)	**0.005**	**0.01**
∆ 4-Hydroxyphenyllactic (1–0)	0.8 (0.3–1.3)	**0.6** (**0.3**–**0.9**)	**1.1** (**0.6**–**1.7**)	**1.1** (**0.6**–**1.8**)	**0.001**	**<0.001**
Σ3AMM (0)	2.3 (1.9–3.1)	2.3 (1.8–3.1)	2.4 (1.9–3.1)	2.6 (1.9–3.3)	0.5	0.6
Σ3AMM (1)	3.4 (2.5–4.8)	**2.7** (**2.3**–**4.0**)	**4.1** (**3.0**–**5.1**)	**4.3** (**3.0**–**5.7**)	**0.001**	**0.002**
∆ Σ3AMM (1–0)	0.9 (0.2–1.8)	**0.6** (**0.1–1.0**)	**1.3** (**0.7–2.4**)	**1.4** (**0.5**–**2.9**)	**0.001**	**0.004**

*** Chi-square test.

**Table 3 biomedicines-11-01335-t003:** The ROC analysis results for the 4-hydroxyphenyllactic acid and Σ3AMM concentrations in serum samples collected 6 h after the end of the surgery (1 point) as the predictors of the postoperative complications in cardiac surgery patients.

Parameter		4-Hydroxyphenyllactic Acid (1)	Σ3AMM (1)
Area Under the Curve		0.686	0.717
Standard Error		0.060	0.058
*p*-Value		0.005	0.001
Asymptotic 95% CI	Lower Bound	0.569	0.604
	Upper Bound	0.804	0.830
Cut-Off Value		2.0 µmol/L	2.9 µmol/L
Sensitivity (95% CI), %		79 (64–90)	81 (67–92)
Specificity (95% CI), %		47 (30–65)	56 (38–72)
Positive Predictive Value, %		64	69
Negative Predictive Value, %		65	71
Accuracy, %		65	70
Odds Ratio		3.4 (1.3–9.0)	5.5 (1.9–15.0)

## Data Availability

Not applicable.

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
