# Peer review of "Microbiota Metabolism Failure as a Risk Factor for Postoperative Complications after Aortic Prosthetics"

_biomedicines, 2023, doi:10.3390/biomedicines11051335_

Round 1
Reviewer 1 Report
In the manuscript “Microbiota Metabolism Failure as a Risk Factor for Postoperative Complications After Aortic Prosthetics”, the authors proposed to study whether the development of postoperative complications in patients with aortic aneurysm was related with initial or acquired disorders of microbiota metabolism. The authors approached this aim by monitoring the level of some aromatic microbial metabolites (AMMs) circulating in the blood before the surgery and in the early postoperative period. The paper could be of interest but there are too many drawbacks that hampered the initial enthusiasm. The data is too preliminary and discussion is absent. I have some specific comments that may be useful.
Specific comments:
1. Patients anthropometric parameters should be included. What is the role of age and the metabolic status of the patients?
2. The data is a bit preliminary and that should be reflected on the title, abstract and discussion.
3. Discussion is very weak. Please identify the gap in the literature and provide clear evidence of the contribution of the data. The rationale is not present and explained. Finally, there is no clear discussion. Study limitations must be included and well addressed. There is no take home message and future perspectives.
4. References must be updated.
Language can be improved.
Author Response
Reviewer 1
In the manuscript “Microbiota Metabolism Failure as a Risk Factor for Postoperative Complications After Aortic Prosthetics”, the authors proposed to study whether the development of postoperative complications in patients with aortic aneurysm was related with initial or acquired disorders of microbiota metabolism. The authors approached this aim by monitoring the level of some aromatic microbial metabolites (AMMs) circulating in the blood before the surgery and in the early postoperative period. The paper could be of interest but there are too many drawbacks that hampered the initial enthusiasm. The data is too preliminary and discussion is absent. I have some specific comments that may be useful.
Specific comments:
- Patients anthropometric parameters should be included. What is the role of age and the metabolic status of the patients?
We thank the expert reviewer for highlighting this issue and allowing us to improve the manuscript. We analyzed the relationship of Σ3AMM (1) with age and Σ3AMM (1) with BMI of included patients (Fig. 1 A-B in Suppl.2). We found a trend towards an increase in Σ3AMM with increasing age that did not reach statistical significance (p = 0.065) and there was no significant relationship between BMI and Σ3AMM (p = 0.892) (lines 236-241, 291-293).
- The data is a bit preliminary and that should be reflected on the title, abstract and discussion.
The data is really a bit preliminary and we have mentioned it in the abstract (line 15) and in discussion (lines 310-335).
- Discussion is very weak. Please identify the gap in the literature and provide clear evidence of the contribution of the data. The rationale is not present and explained. Finally, there is no clear discussion. Study limitations must be included and well addressed. There is no take home message and future perspectives.
We thank the expert reviewer for highlighting this issue and allowing us to improve the manuscript. We extend the Discussion section and add the information about cardiovascular complications and the role of the microbiota (lines 259-274):
“Despite the great progress in cardiovascular surgery, the percentage of postoperative complications, such as infections at the site of surgery, pneumonia and others, does not decrease and, according to the literature, on average is registered from 12.6 to 21%. The studies on this topic provide analyses of the risk factors for complications, which most often included age, gender, chronic lung diseases, heart failure, duration of cardiopulmonary bypass [21–23]. In recent years, there has been a growing interest in studying the potential connection of human microbiota with cardiovascular diseases [24,25] and the role of the intestinal microbiota in the pathogenesis of heart failure [26,27]. Scientific data con-firm the potential involvement of microbiota-related factors, for example, taxonomy features of gut microbiota in cardiosurgical intensive care patients [28], increased risk of bacterial translocation followed by a pro-inflammatory condition [24,29]. The studies consider the possibilities of influencing the microbiota in order to improve the results of treatment of patients with heart diseases in the future [30–32]. However, we did not find any studies describing the relationship between the metabolic activity of the microbiota though metabolites of tyrosine and phenylalanine and the development of cardiac surgical complications.”
Also, we add the limitations, general importance of the results and future perspectives (lines 310-335):
“Our study is a pilot single center study with obvious limitations. In the medical center where the study was conducted, 1087 different cardiac surgical interventions were per-formed in 2021, including 166 (15%) surgeries for aortic aneurysm/dissection. In accordance with the Protocol of the study, patients (n = 79) which were operated by one surgical team in 2021, were analyzed in our study. We purposefully selected operations on the aorta, due to the greatest risk of postoperative complications, taking into account the severity of this pathology and a complex of concomitant diseases in every patient included in the study (Supplementary 1). All patients with this pathology need individually selected options for reconstructive aortic surgery, differing in type, localization, duration, volume, which were listed in Section 3.1. Despite the small number of patients, a disorder in metabolic function of the microbiota as an important risk factor for postoperative complications was identified, which was not previously taken into account. Also, it was found that this risk factor can be determined with a high degree of confidence early enough after surgery (6 hours). The authors assume that this important result will be confirmed with other types of operations in further extensive studies.
It is important to take into account that operations on the aorta in such patients with multiple risk factors are not performed in all cardiac surgery clinics, which makes it difficult to organize multicenter studies, and it is almost impossible to select a group of similar patients.
The authors believe that the results of this pilot study are extremely important for understanding the mechanisms of complications. Impaired microbiota metabolism is important in the development of complications after complex reconstructive aortic surgery, which is the basis for the search for a new prevention strategy. Already today, a randomized clinical trial has begun in the same cardiac surgery center to study the effectiveness of antimicrobial prevention based on the regulation of the metabolic activity of the microbiota.”
- References must be updated.
The additional 13 references on the topic were added in the Discussion section and the reference list was updated.
Reviewer 2 Report
The authors of the present study aimed at investigating the relationship between the development of postoperative complications in patients with aortic aneurysm and initial or acquired disorders of microbiota metabolism, by monitoring the level of some aromatic microbial metabolites (AMMs) circulating in the blood before and after surgery. The study included 79 patients with aortic aneurysm, 36 of whom did not experience complications, and 43 who did. Serum samples were collected before and 6 hours after the surgery. The study found that the sum of three sepsis-associated AMMs was higher before the surgery in comparison to healthy volunteers, and it was also higher in the early postoperative period in patients with all types of complications compared to those without complications. The study suggests that impaired microbiota metabolism may play a role in the development of complications after complex reconstructive aortic surgery and calls for the search for new prevention strategies. Overall, the text is clear and concise.
However, it does not include a paragraph explicitly discussing the limitations of the study. A limitations section is important in any research study as it highlights areas where the research could be improved or expanded upon and provides context for interpreting the findings. Some potential limitations that could be included in a limitations section for this study might be the small cohort, the lack of information on recent infections/antibiotic treatment in the patient cohort, the possibility of confounding factors that were not accounted for, the potential for bias in patient selection, and the possibility of measurement error in the microbiota metabolites.
As mentioned, one limitation of the study is its relatively small cohort size, with only 79 patients with aortic aneurysm included in the study. As a result, it is important to interpret the findings of this study with caution, and further research with larger cohorts should be conducted to confirm these findings and determine the extent to which they apply to a broader population.
Another limitation of the study is the potential heterogeneity of the patient cohort included. It is mentioned that patients with aortic aneurysm were included regardless of the type of surgery they underwent, which could have led to variations in patient outcome and influenced the results.
The text does not mention whether the study included data on the recent history of infections in the patient cohort. This is a potential limitation, as recent infections or antibiotic use could have influenced the composition and function of the microbiota and, consequently, the development of postoperative complications. In addition, infections can be an important factor in the development of aortic aneurysms, and a history of infections could influence the baseline level of AMMs in the patients. Therefore, not accounting for recent infections could limit the study's ability to draw conclusions about the specific role of microbiota metabolism in the development of complications after aortic aneurysm surgery.
By acknowledging these limitations, researchers can better contextualize their findings and make suggestions for future research to improve upon the study's weaknesses.
Author Response
We thank the expert for his positive review allowing us to improve the manuscript. We acknowledge that this is a retrospective single center pilot study with obvious external validity limitations. We extend the Discussion section and add the limitations (lines 310-335):
“Our study is a pilot single center study with obvious limitations. In the medical center where the study was conducted, 1087 different cardiac surgical interventions were performed in 2021, including 166 (15%) surgeries for aortic aneurysm/dissection. In accordance with the Protocol of the study, patients (n = 79) which were operated by one surgical team in 2021, were analyzed in our study. We purposefully selected operations on the aorta, due to the greatest risk of postoperative complications, taking into account the severity of this pathology and a complex of concomitant diseases in every patient included in the study (Supplementary 1). All patients with this pathology need individually selected options for reconstructive aortic surgery, differing in type, localization, duration, volume, which were listed in Section 3.1. Despite the small number of patients, a disorder in metabolic function of the microbiota as an important risk factor for postoperative complications was identified, which was not previously taken into account. Also, it was found that this risk factor can be determined with a high degree of confidence early enough after surgery (6 hours). The authors assume that this important result will be confirmed with other types of operations in further extensive studies.
It is important to take into account that operations on the aorta in such patients with multiple risk factors are not performed in all cardiac surgery clinics, which makes it difficult to organize multicenter studies, and it is almost impossible to select a group of similar patients.”
Also, we added information about antimicrobial treatment in the Section 3.1 (lines 182-188):
“For perioperative antibiotic prophylaxis in 57 (72%) patients, cefazolin was standard-ly used (2 g once immediately before surgery), then it was prescribed at a dosage of 2 g 3 times a day for two or three days. Other cephalosporins (cefuroxime, ceftazidime), or antimicrobials of other groups, or their combinations with vancomycin, fluconazole, etc., were rarely used according to individual indications, taking into account the anamnesis. When comparing indicators in groups of patients with and without complications, who received different antimicrobials for antibiotic prophylaxis, no differences were obtained.”
Round 2
Reviewer 1 Report
The revised version of the manuscript improved the quality
It is acceptable